# Might Dog Walking Reduce the Impact of COPD on Patients’ Life?

**DOI:** 10.3390/healthcare10112317

**Published:** 2022-11-18

**Authors:** Ilaria Baiardini, Salvatore Fasola, Chiara Lorenzi, Nicole Colombo, Matteo Bruno, Stefania La Grutta, Carla Scognamillo, Fulvio Braido

**Affiliations:** 1Respiratory Unit for Continuity of Care, IRCCS, Ospedale Policlinico San Martino, Department of Internal Medicine (DiMI), University of Genova, 16126 Genova, Italy; 2Institute of Translational Pharmacology, National Research Council, 90146 Palermo, Italy; 3Medical Affairs Boehringer Ingelheim, 20139 Milan, Italy

**Keywords:** COPD, dog walking, exacerbation, physical activity

## Abstract

Low levels of physical activity (PA) lead to a worsening of physical condition and contributes to multimorbidity in Chronic Obstructive Respiratory Disease (COPD). Unsupervised PA related to dog ownership may contribute to reducing sedentary behavior. We aimed to investigate the relationship between dog walking, patient-reported outcomes (PROs) and exacerbations in COPD. A pre-defined sample of 200 COPD patients (dog owners and non-dog owners) with symptomatic COPD was sourced from a database representative of the Italian population. A computer-assisted personal interview was used to assess health status impairment (CAT), fatigue (FACIT), health-related quality of life (HRQoL) (EQ-5D), and PA frequency. In the whole sample, PA was associated with better CAT, EQ-5D, VAS, FACIT scores and reduced number of exacerbation (*p* < 0.001). Under the same CAT scores, dog-walking duration was associated with a better HRQoL (EQ5D, *p* = 0.015) and less fatigue (FACIT, *p* = 0.017). In an adjusted regression model, walking dogs >30 min was associated with lower fatigue (FACIT) than having no dogs and walking dogs <15 min (*p* = 0.026 and *p* = 0.009, respectively). Motivation related to dog walking could modify patients’ tendency to focus on symptoms during PA and, therefore, to perceive the fatigue. Dog walking may be effective for increasing and maintaining regular PA, reducing the subjective impact of COPD.

## 1. Introduction

The difficulty in keeping active is a common feature of Chronic Obstructive Respiratory Disease (COPD) depending on symptoms of breathlessness, fatigue and muscle deconditioning [1,2]. It begins in the early stages of the disease and gradually increases over time to a greater extent than in non-COPD subjects [3,4]. Lower levels of PA leads to a further worsening of the physical condition, lung function decline, and contributes to multimorbility [5]. This leads to a downward spiral of inactivity that impairs health outcomes and represents an independent risk factor for COPD-related hospital admissions and worst prognosis [6,7,8,9,10].

Physical activity (PA), defined as “any bodily movement produced by skeletal muscles resulting in an increase in energy expenditure of the body” [11], refers to the overall level of PA carried out by a person at work, at home, for commuting, and during leisure-time. The World Health Organization (WHO) 2020 recommendations [12] are to perform at least 150 min per week of moderate-intensity PA, or at least 75–150 min of vigorous-intensity PA, or an equivalent combination of moderate- and vigorous-intensity activity. The benefits of following these recommendations have been well established: PA is associated with a lower all-cause mortality and cardiovascular disease, morbidity, and disability [13,14]. In addition, there is increasing evidence that engaging in regular PA is associated with better patient-reported outcomes (PROs) such as health-related quality of life (HRQoL), symptoms, and mood [12,15].

However, despite converging evidence for the beneficial effects of being active, it is often challenging to increase engagement with PA in people with COPD [2,16]. Key barriers to maintain exercise behaviors are disease-specific problems (i.e., symptoms, functional limitations, comorbidities) [17,18], psychological factors (i.e., depression, disease-specific anxiety, fear avoidance behaviors [19,20], health attitudes toward fitness and strength [21], lack of motivation [16], shame [22]), in addition to practical difficulties (i.e., lack of transportation) [2] and limited access to interventions for improving PA [23].

Some of these barriers might be overcome through activities that can be easily integrated in individuals’ daily routine, without supervision, and using resources that are available at home (i.e., walking, climbing the stairs, exercises using water bottles as weights).

Recently, a systematic review and meta-analysis, highlighted that unsupervised PA interventions in people with COPD have a positive impact on dyspnea and exercise capacity; such interventions are safe and show a high adherence rate [24]. 

Unsupervised PA related to dog ownership may also contribute to reduce sedentary behavior. People owning dogs have been observed to be more physically active that non-owners [25,26] and, therefore, they are more likely to meet the recommended level of 150 min per week [26,27] and to engage in PA during leisure time [27]. 

The association between dog ownership and increased PA, primarily through dog walking, has been also confirmed in subjects with diabetes [28], cardiovascular diseases [29], breast cancer [30], in chronic hemodialysis patients [31] and in obese people who have undergone a gastric banding procedure [32]. 

Exploring if having a dog is associated with outcomes that are relevant to the experience of COPD patients will help to identify new insights into PA interventions in clinical practice.

The aim of this study is therefore to investigate if dog ownership is associated with PROs (such as health status, HRQoL and fatigue) and exacerbations in patients with COPD.

## 2. Materials and Methods

A cross-sectional survey was carried out between 7–18 March 2019. Trained research staff administered the survey using a computer-assisted personal interviewing (CAPI), a technique employed for data collection on a portable device. The study sample was sourced from the Doxa Population Panel, a proprietary quality-checked database representative of the general Italian population on several key socio-demographic variables. All procedures were in accordance with both international (ESOMAR and EphMRA) and national (FarmIndustria) ethical standards as well as with the 1964 Helsinki Declaration, its later amendments or comparable ethical standards. According to Italian law, when anonymous surveys are conducted without the use of clinical data, ethics approval from the IRB/local ethics committee is not required. Informed consent was obtained from all individual participants involved in the present study.

A pre-defined sample of 200 patients was recruited from a quality-checked database representative of the general Italian population. Inclusion criteria were symptomatic COPD and a COPD Assessment Test (CAT) [33] ≥ 10 (the threshold indicated by GOLD) [34]. Non exclusion criteria have been considered. 

The CAPI system was used to administer the following PROs at the responder home:CAT [33] an 8-item unidimensional measure of health status impairment in COPD. The score ranges from 0 to 40, with higher scores representing worse health.EuroQol-5D (EQ-5D), developed in 1990, is a most widely used generic questionnaire to assess HRQoL [35]. It is applicable to the general population as well as a wide range of health conditions including COPD [36]. It consists of five questions assessing whether subjects were experiencing problems (no, some/moderate, or severe/extreme) in 5 dimensions of health (mobility, self-care, usual activities, pain/discomfort and anxiety/depression). It also includes a vertical visual analogue scale (VAS) asking subjects to rate their overall health on a scale from 0 (the worst imaginable health) to 100 (the best imaginable health).Functional Assessment of Chronic Illness Therapy Fatigue Scale (FACIT-Fatigue), a 13-item questionnaire to assess fatigue and its impact on daily activities and functioning [37], which has been previously used in COPD [38,39,40]. The total score ranges from 0 to 52, with higher scores indicating less fatigue [41].

Socio-demographic characteristics (age, gender, education, employment status, family support), clinical features (CAT score and exacerbations defined as an cute worsening of respiratory symptoms that result in additional therapy, according to GOLD document) [34], self-reported frequency of PA, and dog-walking duration were recorded using an ad-hoc questionnaire. 

Subject characteristics were summarized through the mean and standard deviation (SD) for quantitative variables and through absolute (percentage) frequencies for categorical variables. Comparisons among groups were carried out using one-way ANOVA (which reduces to the t-test in the case of comparison between two groups) for quantitative variables and the chi-square test for categorical variables. 

An exploratory analysis was carried out to compare the distribution of questionnaire scores (CAT, EQ5D, VAS, and FACIT) and exacerbations in the last year (>1 vs. 1) by physical activity frequency (categorized as “never/hardly ever”, “<1 times/week”, “1–2 times/week”, “3–4 times/week”, “almost every day”), dog-ownership duration (“non-dog owner”, “0–2 years”, “3–5 years”, “>5 years”), and dog-walking duration (“non-dog owner”, “<15 min”, “15–30 min”, “>30 min”). For significant associations, pairwise comparisons between groups were carried out using the Holm’s method for addressing the aspect of multiplicity. Non-significant associations were not investigated further in subsequent analyses.

Linear regression models were estimated to formally investigate the association between dog-ownership categories (dog-ownership, dog-ownership duration, and dog-walking duration) and questionnaire scores. Both unadjusted models and models adjusted for age, gender, education level (lower than or at least 8 years), occupational status (retired or not), cohabitation status (living alone or not), and physical activity frequency were estimated. Since the dog-ownership duration and the dog-walking duration were structurally missing in non-dog owners, ad-hoc dummy variables were used if these variables were both included in regression models [42]. Similarly, logistic regression models were estimated using the frequency of exacerbations in the last year (>1 vs. 1) as the outcomes. Associations were reported as mean differences (β coefficients) and 95% confidence intervals for linear regression, and odds ratios (OR) and 95% confidence intervals for logistic regression. Since the distribution of EQ5D was negatively skewed, bootstrap confidence intervals were derived in the relevant linear regression model.

Analyses were carried out using the R statistical software, version 4.0.2 (R Foundation for Statistical Computing, Vienna, Austria). Statistical significance was set at *p* < 0.05.

## 3. Results

A total of 200 subjects were included in this study, of which 99 were dog-owners and 101 were not (Table 1). On average, dog-owners were 3 years younger than non-dog owners (*p* = 0.038). About 50% of the subjects were females, 44% were retired, and 16% were living alone. 

Physical activity frequency was not significantly associated with dog ownership (*p* = 0.071) despite somewhat larger frequencies of classes “3–4 times/week” and “almost every day” observed among dog-owners (18% and 27%, respectively). Mean dog-ownership duration was >5 years for 50% of dog-owners. Dog-walking duration was more than 30 min for 13% of dog-owners.

Questionnaire scores and the frequency of exacerbations were significantly associated with PA frequency in the whole sample (Table 2). Subjects who reported doing PA never/hardly ever had a higher mean CAT score and lower mean EQ5D, VAS, and FACIT scores than other physical activity groups (*p* < 0.001) (Table 2). Subjects who reported doing PA never/hardly ever or <1 times/week had more frequently multiple (>1) exacerbations in the last year than in other groups (*p* < 0.001) (Table 2). Dog-ownership duration was not associated with questionnaire scores and disease outcomes. No difference in CAT score emerged among non-dog owners and dog-owners walking for <15, 15–30, and 30 min. Dog-walking duration was significantly associated with EQ5D (*p* = 0.015) and FACIT scores (*p* = 0.017) (Table 2). In particular, mean scores were the lowest in subjects walking dogs for less than 15 min. Questionnaire scores were similar between non-dog owners and dog owners walking dogs for 15–30 min, while scores were the highest in subjects walking dogs for more than 30 min (Table 2, Figure 1). 

Following this exploratory analysis, dog-ownership duration was not included in regression models. Physical activity frequency was categorized as “physical activity” and “no physical activity” (“never/hardly ever”) in linear regression models (Table 3), and as “regular physical activity” (at least once a week) and “others” in logistic regression models (Table 4). For dog-walking, “>30 min” was used as the reference (Table 3 and Table 4).

In unadjusted models, having no dogs was associated with lower EQ5D (β = −0.09, *p* = 0.043) an FACIT (β = −7.64, *p* = 0.011) scores than walking dogs >30 min, while walking dogs <15 min was associated with lower EQ5D (β = −0.14, *p* = 0.004) an FACIT (β = −9.89, *p* = 0.002) scores than walking dogs >30 min (Table 3). After adjusting PA and other potential confounders, the aforementioned EQ5D differences were attenuated, and having no dogs was not any more associated with a significantly lower EQ5D than walking dogs >30 min. Female gender was associated with lower VAS scores (β = −6.63, *p* < 0.001). Retired subjects and those living alone had somewhat worse questionnaire scores. Physical activity was significantly associated with better questionnaire scores (Table 3). In logistic regression models, no significant associations with dog-walking duration were found. Retired subjects were significantly associated with a higher risk of multiple exacerbations in the last year. Regular physical activity was significantly associated with a lower risk of multiple exacerbations in the last year (Table 4).

## 4. Discussion

Dog walking has been proposed as a purposeful and feasible opportunity for dog owners seeking to maintain regular PA [25,43]. At present, the few available data in patients with COPD have identified dog walking as a socio-environmental factor related to PA [44]. In this study, we assessed the relationship of dog ownership with PROs, self-reported PA and exacerbations in a pre-defined sample of subjects with moderate COPD. 

Our first finding is that having a dog is not significantly associated to the level of PA. Second, PA frequency of COPD patients was significantly associated with questionnaire scores and frequency of exacerbations in the whole sample. Third, a significant association of dog-walking duration with PROs and exacerbations was found. 

The positive association between dog ownership and levels of PA, detected in various populations [28,43], is only a non-significant tendency in our sample, although we found a higher percentage of subjects doing regular PA (3–4 times/week or almost every day) in dog owners (45% vs. 27%). Indeed, dog ownership may not modify habits by itself, as observed in a previous study assessing changes in PA following dog acquisition [45]. This may depend on factors related with both patients (i.e., comorbidities) and dogs (i.e., breed, size, age, health status) that we have not evaluated.

Our results are in line with previous studies that have highlighted the benefits of PA, assessed by a self-report measure, on PROs both in the general population and in patients with chronic diseases and disabilities [46,47,48]. In fact, the effect of PA is also perceived by patients with COPD in terms of health status, HRQoL, and fatigue. Overall, these results indicate that, for patients who spend time in physical exercise, the impact of the disease from a subjective viewpoint is less severe. PA levels were also associated with a higher number of exacerbations, in line with previous studies that have identified physical inactivity as a risk factor in the exacerbation of COPD [40,41,42,43,44,45,46,47,48,49,50,51]. Some mechanisms underlying the potential beneficial impact of PA on exacerbations may be hypothesized. One possible explanation is that a better conditioned cardiovascular system would fit better to the increased oxygen intake in respiratory muscles during COPD exacerbation [52]. Moreover, a reduction of induced lactic acidosis and improvement of the muscular oxidative capacity would lead muscles to better tolerate a COPD exacerbation [53]. Finally, the anti-inflammatory and anti-oxidant effects of PA [54] should have a role.

An interesting result of this study was the positive association of dog-walking duration with better HRQoL and less fatigue, regardless of the extent to which COPD affects patients’ lives. In fact, although non-dog owners and dog-owners walking for <15 and 15-min have no significant difference in CAT scores, subjects walking dogs for more than 30 min have better EQ5D and FACIT scores (Table 2). These results suggest that, even if significant COPD symptoms are present, it is possible to maintain regular activity and to improve HRQoL and fatigue as a function of PA levels.

Linear regression models confirmed the results of the exploratory analysis, highlighting the potential benefits of dog walking in patients with COPD. In unadjusted models, walking dogs >30 min was associated with higher HRQoL (EQ5D) and lower fatigue (FACIT) than having no dogs and walking dogs <15 min. 

After adjusting for PA, age, gender, education level, occupational and cohabitation status, HRQoL differences were attenuated. Conversely, also in adjusted models, walking dogs >30 min was associated with a significantly lower fatigue than having no dogs and walking dogs <15 min. In this regard, it may be assumed that motivation plays a role in perception of effort. According to the psychobiological model of endurance performance based on motivational intensity theory [55], the experience of fatigue may be explained as a form of task disengagement rather than just as a worse COPD status. Dog owners who regularly walk their dog for more than 30 min could be motivated by strong attachment and responsibility toward the dog, as previously found in community samples [56,57] or by considering dog walking as an enjoyable activity. Such motivations may have modified patient’s tendency to focus on symptoms during PA and, therefore, to perceive the fatigue. 

Moreover, in line with previous research, female gender [58,59] and retirement [60] were associated with lower overall HRQoL scores and suggest that health managers and clinicians should consider these features in the management of COPD with the ultimate aim of meeting the specific needs of their patients and increasing their HRQL. A significant association with living condition has been found for CAT and FACIT: in patients who live alone, PA (both in dog owners and non-owners) was significantly associated with worse PROs scores. 

To the best of our knowledge, the relationship between PA, PROs and exacerbations, already known in other diseases, had never been investigated in COPD. However, there are also several limitations that have implications for future research and for the interpretation of our findings. First, it is a retrospective observational study carrying out the limitations of this kind of data, and the results should be considered as hypothesis-generating. Secondly, an objective evaluation of PA was difficult to obtain due to the study design that is based on patient reported outcomes. All information are based on self-reports which may be influenced by potential recall bias: the possibility that data could be under or over reported should be taken into account. The availability of devices to monitor PA and the increase in exercise tolerability may be used in longitudinal studies to obtain objective data. Thirdly, data and adherence concerning specific types of inhaler medications were not available; nor was a more complete staging of COPD. Finally, this study presents limitations in the generalizability of the results to populations outside of Italy.

## 5. Conclusions

In conclusion, the study results highlight the potentially important role that regular PA with a dog could play in reducing the burden of COPD on a patient’s life. Promoting dog walking among dog owners who do not routinely walk their dogs may be an effective strategy for increasing and maintaining regular PA and, consequently, for reducing the impact of COPD on clinical and patient-reported outcomes. Our results should be useful to develop and disseminate public education campaigns to promote PA in COPD patients. Moreover, the pet industry (food, treats, wellness, health) may identify COPD patients as a new target.

## Figures and Tables

**Figure 1 healthcare-10-02317-f001:**
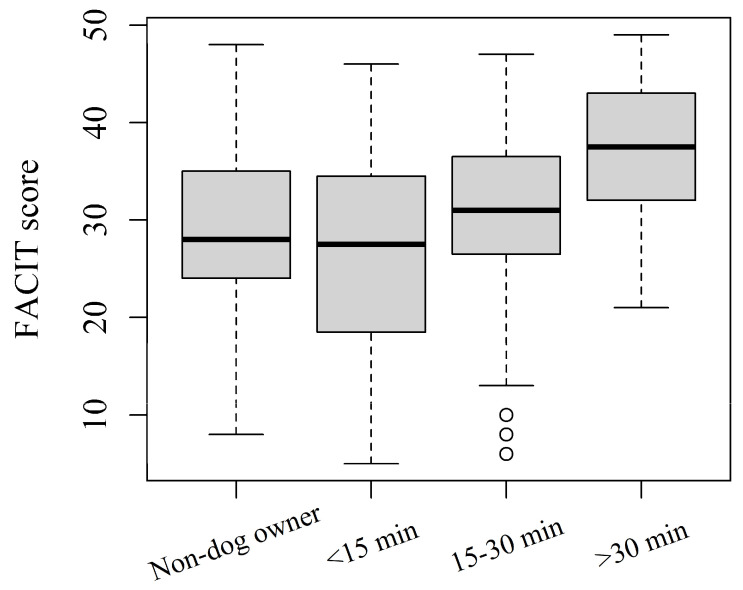
The distribution of FACIT score by dog-walking duration. Boxplots represent the median (central line), 25th–75th percentiles (box), and min-max non-outlier values (whiskers).

**Table 1 healthcare-10-02317-t001:** Participant characteristics by dog-ownership.

	Overall *n* = 200	Dog Owners *n* = 99	Non-Dog Owners *n* = 101	*p*-Value ^1^
Age (years)	64.35 (9.56)	62.9 (9.4)	65.7 (9.6)	**0.038**
Gender				1.000
Male	102 (51)	50 (51)	52 (51)	
Female	98 (49)	49 (49)	49 (49)	
Education ≥8 years	132 (66)	69 (70)	63 (62)	0.345
Retired	89 (44)	42 (42)	47 (47)	0.658
Living alone	31 (16)	13 (13)	18 (18)	0.471
Physical activity frequency				0.071
Never/hardly ever	63 (32)	27 (27)	36 (36)	
<1 times/week	29 (14)	11 (11)	18 (18)	
1–2 times/week	36 (18)	16 (16)	20 (20)	
3–4 times/week	26 (13)	18 (18)	8 (8)	
Almost every day	46 (23)	27 (27)	19 (19)	
Dog-ownership duration				-
Non-dog owner	101 (50)	0 (0)	101 (100)	
0–2 years	14 (7)	14 (14)	0 (0)	
3–5 years	34 (17)	34 (34)	0 (0)	
>5 years	51 (26)	51 (52)	0 (0)	
Dog-walking duration				-
Non-dog owner	101 (50)	0 (0)	101 (100)	
<15 min	40 (20)	40 (40)	0 (0)	
15–30 min	47 (24)	47 (47)	0 (0)	
>30 min	12 (6)	12 (13)	0 (0)	

Data are presented as *n* (%) or mean (SD). ^1^ One-way ANOVA for quantitative variables, chi-square test for categorical variables. Significant *p*-values (<0.05) are in bold.

**Table 2 healthcare-10-02317-t002:** The distribution of questionnaire scores (mean, SD) and exacerbations (*n*, %) by physical activity, dog-ownership duration, and dog-walking duration.

Physical Activity	Never/Hardly Ever (1)	<1 Times/Week (2)	1–2 Times/Week (3)	3–4 Times/Week (4)	Almost Every Day (5)	*p*-Value ^1^	Group Separation ^2^
CAT	29.32 (5.35)	23.17 (7.27)	24.42 (5.13)	24.15 (5.24)	23.93 (5.80)	**<0.001**	{2543}{1}
EQ5D	0.69 (0.20)	0.83 (0.13)	0.81 (0.11)	0.84 (0.08)	0.85 (0.07)	**<0.001**	{1}{3245}
VAS	53.97 (14.19)	64.45 (13.68)	61.42 (13.45)	62.69 (10.29)	64.98 (10.50)	**<0.001**	{1}{3425}
FACIT	23.27 (8.75)	32.72 (10.99)	31.89 (7.89)	32.54 (9.53)	31.96 (8.62)	**<0.001**	{1}{3542}
>1 exacerb.	35 (56)	13 (45)	8 (22)	5 (19)	12 (26)	**<0.001**	{4352}{21}
Dog-ownership	Non-dog owner (1)	0–2 years (2)	3–5 years (3)	>5 years (4)	-	*p*-value ^1^	Group separation ^2^
CAT	25.16 (6.31)	24.86 (8.24)	25.32 (5.35)	27.00 (5.87)	-	0.340	-
EQ5D	0.79 (0.15)	0.83 (0.16)	0.78 (0.17)	0.79 (0.15)	-	0.735	-
VAS	60.45 (14.99)	66.57 (10.19)	60.47 (9.43)	58.94 (13.02)	-	0.316	-
FACIT	29.03 (9.14)	32.57 (14.17)	29.38 (9.87)	29.25 (10.06)	-	0.663	-
>1 exacerb.	41 (41)	3 (21)	10 (29)	19 (37)	-	0.415	-
Dog-walking	Non-dog owner (1)	<15 min (2)	15–30 min (3)	>30 min (4)	-	*p*-value ^1^	Group separation ^2^
CAT	25.16 (6.31)	27.52 (6.55)	25.49 (5.40)	23.92 (6.47)	-	0.154	-
EQ5D	0.79 (0.15)	0.74 (0.20)	0.82 (0.11)	0.88 (0.06)	-	**0.015**	{213}{34}
VAS	60.45 (14.99)	58.42 (12.71)	60.77 (10.47)	66.75 (11.34)	-	0.313	-
FACIT	29.03 (9.14)	26.77 (11.05)	30.55 (9.82)	36.67 (8.81)	-	**0.017**	{213}{34}
>1 exacerb.	41 (41)	15 (38)	14 (30)	3 (25)	-	0.504	-

^1^ One-way ANOVA for quantitative variables, chi-square test for categorical variables. Significant *p*-values (<0.05) are in bold. ^2^ Group separation: group numbers are reported in increasing order of mean/percentage for each variable. After applying the Holm’s method for multiple comparisons, significant separation occurs between groups included within different pairs of brackets. For instance, {213}{34} indicates similarity between pairs 2–1, 2–3, 1–3 and 3–4, and a statistically significant difference between pairs 2–4 and 1–4. CAT: COPD Assessement Test; FACIT-Fatigue: Functional Assessment of Chronic Illness Therapy Fatigue Scale; VAS: visual analogue scale.

**Table 3 healthcare-10-02317-t003:** Questionnaire scores: crude and adjusted β coefficients (mean differences) and 95% confidence intervals from linear regression models.

Unadjusted Models	CAT		EQ5D		VAS		FACIT	
	β	*p*-Value	β	*p*-Value	β	*p*-Value	β	*p*-Value
Intercept	23.92 (20.41, 27.43)	**<0.001**	0.88 (0.85, 0.91) ^§^	**<0.001**	66.75 (59.12, 74.38)	**<0.001**	36.67 (31.15, 42.18)	**<0.001**
Dog-walking >30 min (ref.)	-	-	-	-	-	-	-	-
Non-dog owner	1.24 (−2.47, 4.96)	0.510	−0.09 (−0.14, −0.05) ^§^	**0.043**	−6.30 (−14.38, 1.77)	0.125	−7.64 (−13.47, −1.80)	**0.011**
Dog-walking <15 min	3.61 (−0.39, 7.61)	0.077	−0.14 (−0.22, −0.08) ^§^	**0.004**	−8.33 (−17.03, 0.38)	0.061	−9.89 (−16.18, −3.60)	**0.002**
Dog-walking 15–30 min	1.57 (−2.36, 5.51)	0.431	−0.07 (−0.11, 0.02) ^§^	0.177	−5.98 (−14.54, 2.57)	0.169	−6.11 (−12.29, 0.07)	0.053
**Adjusted Models**	**CAT**		**EQ5D**		**VAS**		**FACIT**	
	**β**	***p*-Value**	**β**	***p*-Value**	**β**	***p*-Value**	**β**	***p*-Value**
Intercept	29.51 (22.16, 36.87)	**<0.001**	0.77 (0.59, 0.97) ^§^	**<0.001**	66.14 (50.44, 81.83)	**<0.001**	14.28 (3.13, 25.43)	**0.012**
Dog-walking >30 min (ref.)	-	-	-	-	-	-	-	-
Non-dog owner	−0.42 (−3.91, 3.07)	0.813	−0.05 (−0.10, 0.01) ^§^	0.262	−2.23 (−9.68, 5.21)	0.555	−6.00 (−11.29, −0.71)	**0.026**
Dog-walking <15 min	1.60 (−2.16, 5.37)	0.402	−0.09 (−0.16, −0.03) ^§^	**0.043**	−3.98 (−12.01, 4.06)	0.330	−7.61 (−13.32, −1.90)	**0.009**
Dog-walking 15–30 min	0.77 (−2.86, 4.41)	0.675	−0.04 (−0.09, 0.01) ^§^	0.348	−3.28 (−11.04, 4.49)	0.406	−4.97 (−10.49, 0.54)	0.077
Age (unit increase)	−0.03 (−0.15, 0.08)	0.584	0.0007 (−0.002, 0.003) ^§^	0.584	−0.02 (−0.27, 0.23)	0.868	0.31 (0.13, 0.49)	**0.001**
Female gender	0.88 (−0.75, 2.51)	0.287	−0.03 (−0.07, 0.01) ^§^	0.098	−6.63 (−10.1, −3.15)	**<0.001**	−2.10 (−4.56, 0.37)	0.095
Education <8 years	0.14 (−1.74, 2.02)	0.883	−0.02 (−0.07, 0.02) ^§^	0.334	−1.75 (−5.77, 2.26)	0.390	−1.2 (−4.06, 1.65)	0.406
Retired	1.08 (−1.09, 3.24)	0.327	−0.06 (−0.10, −0.01) ^§^	**0.021**	−5.75 (−10.37, −1.13)	**0.015**	−4.64 (−7.92, −1.36)	**0.006**
Living alone	2.23 (0.03, 4.43)	**0.047**	−0.05 (−0.12, 0.02) ^§^	0.064	−1.27 (−5.98, 3.43)	0.594	−3.85 (−7.19, −0.51)	**0.024**
Physical activity	−4.92 (−6.72, −3.12)	**<0.001**	0.12 (0.07, 0.17) ^§^	**<0.001**	7.31 (3.46, 11.16)	**<0.001**	7.20 (4.47, 9.94)	**<0.001**

Significant *p*-values (<0.05) are in bold. ^§^ Bootstrap confidence intervals (1000 replications).

**Table 4 healthcare-10-02317-t004:** Exacerbations: crude and adjusted odds ratios and 95% confidence intervals from logistic regression models.

Unadjusted Models	>1 Exacerbations	
	Odds Ratio	*p*-Value
Dog-walking >30 min (ref.)	-	-
Non-dog owner	2.05 (0.57, 9.66)	0.303
Dog-walking <15 min	1.80 (0.45, 9.07)	0.428
Dog-walking 15–30 min	1.27 (0.32, 6.38)	0.744
**Adjusted Models**	**>1 Exacerbations**	
	**Odds Ratio**	***p*-Value**
Dog-walking >30 min (reference)	-	-
Non-dog owner	1.31 (0.32, 6.81)	0.725
Dog-walking <15 min	1.02 (0.22, 5.71)	0.985
Dog-walking 15–30 min	0.85 (0.19, 4.63)	0.839
Age (unit increase)	0.96 (0.92, 1.01)	0.123
Female gender	1.42 (0.75, 2.68)	0.279
Education <8 years	0.70 (0.32, 1.47)	0.347
Retired	2.63 (1.10, 6.56)	**0.033**
Living alone	0.76 (0.31, 1.78)	0.541
Regular physical activity ^§^	0.30 (0.15, 0.57)	**<0.001**

Significant *p*-values (<0.05) are in bold. ^§^ At least once a week.

## Data Availability

With publication, we declare to make the data that support the findings of this study available on request. This should be done through the corresponding author.

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
