# Peer review of "Might Dog Walking Reduce the Impact of COPD on Patients’ Life?"

_healthcare, 2022, doi:10.3390/healthcare10112317_

Round 1

Reviewer 1 Report

The aim of the present study was to investigate whether dog ownership, health factors, and exacerbation where related.

Introduction:

__a: It is not yet clear to me why these 3 aspects are put in relation? Please derive a little more precisely and name the presumed links.

Methods:

__a: The setting of the study and the data collection is not clearly described. What were the eligibility criteria, and the sources and methods of selection of participants? Explain how study size was arrived at. Why n=200 – due to a priori sample size calculation? Consider use of a flow-diagram.

__b: Please clearly define all outcome, exposures and covariables. Give also details of methods of assessment. EQ-5D is not well described. How was PA measured? What exactly does exacerbation mean?

__c: You describe a colorful mix of statistical methods used. You have not yet mentioned the t-test for age in table 1. What about the multiplicity aspect? What were the statistical hypotheses?

__d: Please specify exactly the variables included in the regression models

Results:

__a: Why have differences in Table 1 been statistically tested and reported only for individual variables?

__b: Why were untypical dummy variables formed for the regression models? What is the message behind this? Why were the minutes of walking not compared with those who never walk?

__c: Please report the intercept + ß-coefficient in the linear regression models

__d: Were all conditions for the regression models met?

Discussion:

__a: Despite the methodological issues indicated, the authors should mainly strengthen the discussion. How can the authors state that their results are consistent with those of other PA studies when it is not clear how PA was measured? Can PA of adults with COPD be compared to those with prostate cancer?

__b: The authors state that the main finding of their study is that an interesting result was the positive effect of dog-walking duration with HRQoL and fatigue. Unfortunately, I cannot find this reflected in the results presented.

__c: In my opinion, the major weakness of this study is the use of self-reported PA. The authors should address this limitation in more detail.

__d: Please discuss the generalizability of your study results carefully

__e: Lastly, I think there is a lack of specific conclusions related to health care systems, industry, technology, policy, and regulation.

Author Response

Response to Reviewer 1: Thank you for the time that you spent reviewing the manuscript and providing expert recommendations to strengthen it.

Reviewer 1

The aim of the present study was to investigate whether dog ownership, health factors, and exacerbation where related.

Introduction:

__a: It is not yet clear to me why these 3 aspects are put in relation? Please derive a little more precisely and name the presumed links.

Answer: This suggestion was well-taken. We have improved the introduction to better specify our aim.

Methods:

__a: The setting of the study and the data collection is not clearly described. What were the eligibility criteria, and the sources and methods of selection of participants? Explain how study size was arrived at. Why n=200

– due to a priori sample size calculation? Consider use of a flow-diagram.

Answer: There was no a priori sample size calculation .

However, based on a posteriori calculations, considering for example the conditional means of EQ5D in the four dog-walking groups (Non-dog owner: 0.79; <15 minutes: 0.74; 15-30 minutes: 0.82; >30 minutes: 0.88) and assuming a constant standard deviation of 0.10 and an alpha error of 5%, a sample size of 12 patients per group would have ensured a power of 80% for testing the null hypothesis that there were no differences among the means of the four groups. The actual sample sizes per group were indeed equal to or higher than 12 (Non-dog owner: 101; <15 minutes: 40; 15-30 minutes: 47; >30 minutes: 12) and ensured a power of 99% (https://homepage.univie.ac.at/robin.ristl/samplesize.php?test=anova).

__b: Please clearly define all outcome, exposures and covariables. Give also details of methods of assessment. EQ-5D is not well described. How was PA measured? What exactly does exacerbation mean?

Answer: Method section has been revised to better define outcomes and procedures.

__c: You describe a colorful mix of statistical methods used. You have not yet mentioned the t-test for age in table 1.

Answer: Please, note that the t-test is a special case of one-way ANOVA (the test that we mentioned) when the grouping factor has only two levels. We have now clarified this point in the reviewed manuscript. Thank you for pointing that out.

What about the multiplicity aspect? What were the statistical hypotheses?

Answer: In the exploratory analyses we performed ANOVA/chi-square tests for testing the hypothesis that there were no differences among means/percentages of all groups. The aim of this exploratory analysis was to identify non-significant associations and excluding them from subsequent analyses. Therefore, since we did not perform multiple pairwise comparisons, we did not consider the multiplicity aspect. Thank you.

__d: Please specify exactly the variables included in the regression models

Answer: As we clarified in the Methods section, the variables included in the regression models were dog-ownership categories (dog-ownership, dog-ownership duration, and dog-walking duration), age, gender, education level (lower than or at least 8 years), occupational status (retired or not), cohabitation status (living alone or not), and physical activity frequency. Moreover, we have now added a sentence to provide clarifications about the inclusion of variables with structurally missing answers (see below).

Results:

__a: Why have differences in Table 1 been statistically tested and reported only for individual variables?

Answer: Table 1 provides simple descriptive statistics (n-% or mean-SD) for each individual variable, both overall and stratified by dog-ownership. Accordingly, group differences were also tested for each variable individually.

__b: Why were untypical dummy variables formed for the regression models? What is the message behind this? Why were the minutes of walking not compared with those who never walk?

Answer:

Please note that, as we have now clarified in the Methods section, since the dog-ownership duration (finally not included according to the results of the exploratory analysis) and the dog-walking duration were structurally missing in non-dog owners, we used ad-hoc dummy variables to define the level contrasts applied in regression models, according to the methodology described in “Lipovetsky, S., and E. Nowakowska. 2013. “Modeling with Structurally Missing Data by OLS and Shapley Value Regressions”. International Journal of Operations and Quantitative Management 19:169-178”.

In brief, the method works as follows:

1) choose a reference category for the variable “dog-ownership”, which was “non-dog owner” in our case, and include a dummy variable (say “D1”) that is 1 for “dog owner” and 0 for “non-dog owner”;

2) choose a reference category for the variable “dog-ownership duration”, which was “>30 minutes” in our case, and include two dummy variables:

     - “D2” that is 1 for “15-30 minutes” and 0 otherwise (it is set to 0 also for structurally missing values, i.e. for non-dog owners);

     - “D3” that is 1 for “<15 minutes” and 0 otherwise (it is set to 0 also for structurally missing values, i.e. for non-dog owners).

Please note that, for “dog-ownership duration”, we chose “>30 minutes” as the reference category since we were expecting “15-30 minutes” and “<15 minutes” to be the “risk factors”.

As a result of the above parametrization, as reported in the Results section, the regression coefficient of D1 represents the difference between “>30 minutes” and “non-dog owner”, the regression coefficient of D2 represents the difference between “15-30 minutes” and “>30 minutes”, and the regression coefficient of D3 represents the difference between “<15 minutes” and “>30 minutes”. We thank the Reviewer and hope to have clarified this issue.

__c: Please report the intercept + ß-coefficient in the linear regression models

Answer: In Table 3, we have now added model intercepts and clarified that the “mean differences” are, indeed, the ß-coefficients. Thank you.

__d: Were all conditions for the regression models met?

Answer: Yes, all the conditions were substantially met. A minor concern was raised by the distribution of EQ5D, that was positively skewed. However, applying a log transformation or hypothesize a Gamma distribution (rather than the Gaussian one) was not feasible, since negative (or zero) values are allowed for EQ5D. Additionally, trying to apply these workarounds after restricting to positive values provided substantially the same results as in Table 3.

Discussion:

__a: Despite the methodological issues indicated, the authors should mainly strengthen the discussion. How can the authors state that their results are consistent with those of other PA studies when it is not clear how PA was measured? Can PA of adults with COPD be compared to those with prostate cancer?

Answer: We have reviewed the text and better specified that our study, and the studies we included in the discussion (references 45-47) use a self-report measure of PA. What we want to say is that our results are in line with those of other studies in which PA has been assessed by a self-report measure. In other words, the benefits of self-reported PA on patients experience we found in our sample have also been described in other populations

__b: The authors state that the main finding of their study is that an interesting  result was the positive effect of dog-walking duration with HRQoL and fatigue. Unfortunately, I cannot find this reflected in the results presented.

Answer: Dog-walking duration was significantly associated with EQ5D (p=0.015) and FACIT scores (p=0.017) (Table 2). The discussion has been revised to make more clear this result.

__c: In my opinion, the major weakness of this study is the use of self-reported PA. The authors should address this limitation in more detail.

Answer: We are aware of this limitation and have highlighted it better in the revised version of our manuscript

__d: Please discuss the ++ of your study results carefully

Answer: all the ++ of our results have been discussed

__e: Lastly, I think there is a lack of specific conclusions related to health care systems, industry, technology, policy, and regulation.

Answer: discussion and conclusions have been modified to address reviewer’s suggestions.

Reviewer 2 Report

Interesting study on dog walking and the health of individuals with COPD. Although no association was found between dog owners and PA levels, this study presents relevant results in dog walking and patients health quality and on PA frequency and health scores.

Some considerations:

1) It is not described in detail about the variables collected in the sociodemographic and clinical characteristics questionnaire (line 105).

2) It was mentioned that the tests were performed in the R program, but the packages used in the analysis are not included (line 126).

3) About exploratory analysis details should be in materials and methods (line 152).

4) Include legends in the tables with information about data (abbreviations) and p-value of the test performed.

5) Also include legends in figure 1.

6) In the discussion, it would be interesting to include a brief physiological information about the exacerbations and the frequency of PA (line 195).

7) Associations between female sex and retirement were not discussed (line 221).

Author Response

Reviewer 2

Interesting study on dog walking and the health of individuals with COPD. Although no association was found between dog owners and PA levels, this study presents relevant results in dog walking and patients health quality and on PA frequency and health scores.

Answer: Thank you so much for your comment.

Some considerations:

  • It is not described in detail about the variables collected in the sociodemographic and clinical characteristics questionnaire (line 105).

Answer: sociodemographic and clinical features have been described in detail

2) It was mentioned that the tests were performed in the R program, but the packages used in the analysis are not included (line 126).

Answer: Thank you for this question. Please, note that we did not use any specific package, since regression tools are pre-loaded in the mentioned R version.

3) About exploratory analysis details should be in materials and methods (line 152).

Answer: Please note that all the details provided in the Results section (i.e. decisions about variable categorizations and levels contrasts) strictly follows from the results of the exploratory analysis, and could not be anticipated, without loss of generality, in the Methods section. Conversely, in the Methods section, we limited to specify more in general that: “An exploratory analysis was carried out […]. Similar groups were aggregated, while non-significant associations were not investigated further in subsequent analyses” and that “Since the dog-ownership duration and the dog-walking duration were structurally missing in non-dog owners, we used ad-hoc dummy variables to define the level contrasts applied in regression models”. Thank you.

4) Include legends in the tables with information about data (abbreviations) and p-value of the test performed.

Answer: We have amended legends as suggested, thank you.

5) Also include legends in figure 1.

Answer: We have added legend, as suggested. Thank you

6) In the discussion, it would be interesting to include a brief physiological information about the exacerbations and the frequency of PA (line 195).

Answer: Thank you for the suggestion. Some mechanisms underlying the potential beneficial impact of PA on exacerbations have been discussed.

7) Associations between female sex and retirement were not discussed (line 221).

Answer: Association of PROS to gender and employment status has been discussed

Round 2

Reviewer 1 Report

Dear authors,thank you for revising the manuscript.

Unfortunately, it is still not clear how and where the study participants were recruited. What the inclusion and exclusion criteria were. Why this sample size was achieved. This is relevant information to draw conclusions from the results. Are 200 patients representative of 3.5 million? Is it an ad-hoc (convenience) sample? What do the results mean then?

In addition, the authors do not address the aspect of multiplicity. The authors state that they performed an exploratory analysis; but then please do not provide p-values, only the confidence interval to the estimators.

Also, no speak of significance would be better, and you'd better avoid words like 'impact' and 'effect' in your title and manuscript.

Tables 3 and do not contain confidence intervals. The tables are still difficult to read. I recommend listing the reference category of a variable, e.g. non-dog owner, first, indicating that it is the reference category, then continuing with the categories (dog walking 15-30, >30, ...) in the following rows.

Furthermore, with EQ-5D the recommendation is to calculate SE by bootstrapping. This should be implemented in this way.

Author Response

Unfortunately, it is still not clear how and where the study participants were recruited. What the inclusion and exclusion criteria were. Why this sample size was achieved. This is relevant information to draw conclusions from the results. Are 200 patients representative of 3.5 million? Is it an ad-hoc (convenience) sample? What do the results mean then?

Answer: the method section has been modified to better specify inclusion criteria and sample size.

In addition, the authors do not address the aspect of multiplicity. The authors state that they performed an exploratory analysis; but then please do not provide p-values, only the confidence interval to the estimators.

Also, no speak of significance would be better, and you'd better avoid words like 'impact' and 'effect' in your title and manuscript.

Answer: Please note that the exploratory analysis aimed to identify the significant associations (through the p-value from ANOVA and chi-square tests) to include in the subsequent regression models. Indeed, after the exploratory analysis, we decided not to include dog-ownership in regression models. Moreover, in the revised manuscript, we now addressed the aspect of multiplicity (pairwise comparisons between groups) in Table 2 and updated the Methods section accordingly. Thank you for this comment.

Tables 3 and do not contain confidence intervals.

Answer: Following the Reviewer suggestion, we added confidence intervals in Table 3 and Table 4.

The tables are still difficult to read. I recommend listing the reference category of a variable, e.g. non-dog owner, first, indicating that it is the reference category, then continuing with the categories (dog walking 15-30, >30, ...) in the following rows.

Answer: We modified Table 3 and Table 4 according to the Reviewer suggestion. However, please note that according to the group separations identified by the pairwise comparisons in Table 2, selection of “Dog-walking >30 minutes” as the reference category appears the most informative choice, since it allows to include all the significant contrasts identified (“Dog-walking >30 minutes” vs “Non-dog owner” and “Dog-walking <15 minutes” vs “Non-dog owner”). Thank you for this comment.

Furthermore, with EQ-5D the recommendation is to calculate SE by bootstrapping. This should be implemented in this way.

Answer: Following the Reviewer comment, we used bootstrap confidence intervals in the regression models for EQ5D and updated Table 3 accordingly. Thank you for the suggestion.
